# Multi-grained Correspondence Learning of Audio-language Models for Few-shot Audio Recognition

Shengwei Zhao
National Key Laboratory of Human-Machine Hybrid
Augmented Intelligence, National Engineering Research
Center for Visual Information and Applications, and
Institute of Artificial Intelligence and Robotics, Xi'an
Jiaotong University
Xi'an, Shaanxi, China
zhaoshengwei@stu.xjtu.edu.cn

Linhai Xu
National Key Laboratory of Human-Machine Hybrid
Augmented Intelligence, National Engineering Research
Center for Visual Information and Applications, and
Institute of Artificial Intelligence and Robotics, Xi'an
Jiaotong University
Xi'an, Shaanxi, China
xlh@mail.xjtu.edu.cn

Yuying Liu
National Key Laboratory of Human-Machine Hybrid
Augmented Intelligence, National Engineering Research
Center for Visual Information and Applications, and
Institute of Artificial Intelligence and Robotics, Xi'an
Jiaotong University
Xi'an, Shaanxi, China
lyy0026@stu.xjtu.edu.cn

Shaoyi Du✉
National Key Laboratory of Human-Machine Hybrid
Augmented Intelligence, National Engineering Research
Center for Visual Information and Applications, and
Institute of Artificial Intelligence and Robotics, Xi'an
Jiaotong University
Xi'an, Shaanxi, China
dushaoyi@xjtu.edu.cn

## Abstract

Large-scale pre-trained audio-language models excel in general multi-modal representation, facilitating their adaptation to downstream audio recognition tasks in a data-efficient manner. However, existing few-shot audio recognition methods based on audio-language models primarily focus on learning coarse-grained correlations, which are not sufficient to capture the intricate matching patterns between the multi-level information of audio and the diverse characteristics of category concepts. To address this gap, we propose multi-grained correspondence learning for bootstrapping audio-language models to improve audio recognition with few training samples. This approach leverages generative models to enrich multi-modal representation learning, mining the multi-level information of audio alongside the diverse characteristics of category concepts. Multi-grained matching patterns are then established through multi-grained key-value cache and multi-grained cross-modal contrast, enhancing the alignment between audio and category concepts. Additionally, we incorporate optimal transport to tackle temporal misalignment and semantic intersection issues in fine-grained correspondence learning, enabling flexible fine-grained matching. Our method achieves state-of-the-art results on multiple benchmark datasets for few-shot audio recognition, with comprehensive ablation experiments validating its effectiveness.

✉ Corresponding author.

## CCS Concepts

• **Applied computing** → **Sound and music computing**; • **Computing methodologies** → *Supervised learning by classification.*

## Keywords

Few-shot audio recognition, Multi-grained correspondence learning, Audio-language models, Optimal transport

### ACM Reference Format:

Shengwei Zhao, Linhai Xu, Yuying Liu, and Shaoyi Du✉. 2024. Multi-grained Correspondence Learning of Audio-language Models for Few-shot Audio Recognition. In *Proceedings of the 32nd ACM International Conference on Multimedia (MM '24), October 28-November 1, 2024, Melbourne, VIC, Australia.* ACM, New York, NY, USA, 9 pages. https://doi.org/10.1145/3664647.3681389

## 1 Introduction

Large-scale pre-trained audio-language models [10, 11, 15, 38] demonstrate powerful general multi-modal representation capabilities. However, directly fine-tuning the audio-language model with downstream task data will disrupt the pre-set embedding space. To adapt the audio-language model to downstream tasks, recent work [24, 43] has achieved good performance on few-shot audio recognition by fine-tuning the audio-language model through efficient adapters for coarse-grained correspondence learning. Among them, Treff-adapter [24] mines the correlation between audio and category concepts through coarse-grained key-value cache and zero-shot coarse-grained cross-modal contrast.

The above methods achieve good performance, but their coarse-grained correspondence learning is not sufficient to represent the complex matching patterns between audio and category concepts. Taking the audio of "moving train" as an example, the short whistle sound and the continuous wheel rolling sound are the two corresponding discriminant characteristics. The former requires

attention to audio details (clip-level), while the latter requires attention to the overall audio (overall-level). Coarse-grained audio representation makes it difficult to pay attention to the whole and details of the audio at the same time, and a short category name ("moving train") cannot fully represent the potential discriminative characteristics of the category concept (e.g. whistle, wheel rolling). Therefore, coarse-grained correspondence learning is difficult to establish multi-grained matching patterns between the multi-level information of audio and the diverse characteristics in category concepts.

In order to solve the above problems, we propose to perform multi-grained correspondence learning to better guide the audio-language model to adapt to few-shot audio recognition. Our method contains three core designs: gen-assisted multi-modal representation learning, multi-grained key-value cache and multi-grained cross-modal contrast. Specifically, we first perform multi-modal representation learning with the assistance of the generative models, which is used to mine the multi-level information of audio and the diverse characteristics of category concepts. We then propose multi-grained key-value cache and multi-grained cross-modal contrast for effectively learning multi-grained matching patterns between audio and category concepts. The former maps the categories of test audio by capturing multi-grained correlations between test audio representations and known category audio representations. While the latter mines caption-audio complementary patterns to learn multi-grained correlations between audio fusion representations and category explanation representations.

In addition, we noticed that there are problems of temporal misalignment and semantic intersection when performing fine-grained correspondence learning. Taking the audio of "moving train" as an example, the key information in the audio (whistle sound, rail banging, wheel rolling) can appear at the beginning, end or continuously of the audio. As shown in Fig.1-a, the relevant parts of two similar audios may not appear in the same time segment (temporal misalignment). And as shown in Fig.1-b, some segments in the audio can express multiple semantics at the same time (semantic intersection). In order to deal with the above problems, since optimal transport can establish flexible correspondences between sampling points of different probability distributions, we propose to apply optimal transport to improve fine-grained key-value cache and fine-grained cross-modal contrast. Specifically, we model the empirical distribution of fine-grained audio representation and category explanation representation, and guide flexible fine-grained matching between audio and category by comparing the distance between distributions under the optimal transport framework.

To effectively showcase the efficacy and performance of our proposed method, we meticulously carry out a sufficient number of experiments on two prominent few-shot audio recognition benchmark datasets, namely ESC-50 and FSDkaggle18. When contrasted with the previously existing methods, our proposed multi-grained correspondence learning approach successfully attains state-of-the-art outcomes and demonstrates superior results. We hope that our findings will inspire more research on few-shot audio recognition. Our main contributions can be summarized as follows:

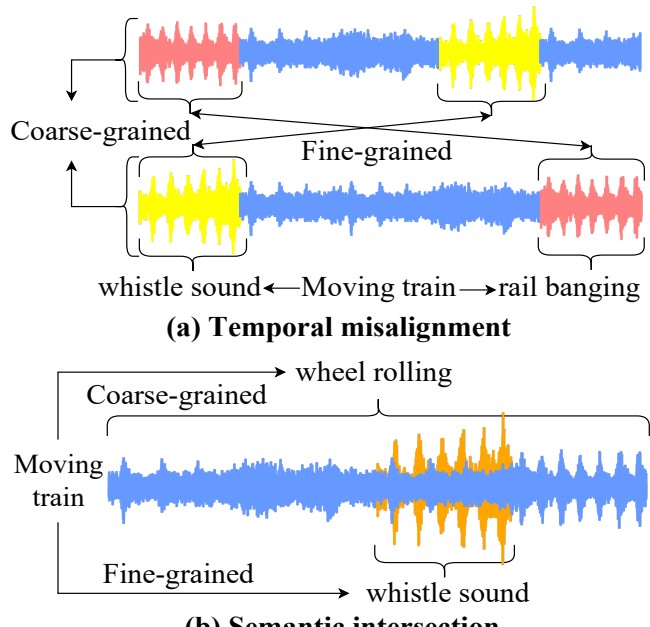

**(a) Temporal misalignment**

**(b) Semantic intersection**

Figure 1: (a) Temporal misalignment: The relevant parts (yellow/red) of two similar audios do not appear in the same time segment. (b) Semantic intersection: Some segments in the audio can express multiple semantics at the same time. The orange part of the audio contains both the sound of a whistle and the sound of rolling wheels.

- We propose multi-modal representation learning assisted by generative models, which effectively mine the multi-level information of audio and the diverse characteristics of category concepts.
- We propose multi-grained correspondence learning to establish multi-grained matching patterns between audio and category concepts through multi-grained key-value cache and multi-grained cross-modal contrast.
- We propose to apply optimal transport to deal with the temporal misalignment and semantic intersection problems existing in fine-grained correspondence learning, and to guide the flexible fine-grained matching.

## 2 RELATED WORK

### 2.1 Audio-Language Models

In recent years, audio-language models [10, 11, 15, 38, 44] have attracted increasing attention due to their powerful universal multi-modal representation capabilities. Taking the representative CLAP [10] as an example, it follows the idea of Contrastive Image-Language Pre-training (CLIP) [32]. Independent audio encoders and text encoders are employed to extract audio representations and their corresponding text representations. Subsequently, contrastive learning is utilized to project both the audio and text representations into a unified embedding space. This process brings matching audio-text pairs closer together in the embedding space while pushing apart

unmatched pairs, thus enhancing the alignment and discrimination between relevant and irrelevant audio-text combinations. With the assistance of training data of large-scale audio-text pairs, audio-language model has good generalization ability to a wide range of downstream audio tasks. By evaluating the similarity between audio and category label names, audio-language models can perform downstream audio recognition without training. However, the gap between pre-training data and downstream task data reduces the generalization performance of audio-language models, which has prompted recent research on guiding audio-language models to adapt to downstream tasks, such as efficient adapter [24, 43] for few-shot audio recognition.

## 2.2 Few-shot Audio Recognition

With the introduction of large-scale audio datasets [9, 12, 13, 18, 28, 30] and the development of audio representation learning theory [2, 3, 14, 27] in recent years, audio classification has made great progress as a benchmark task in the audio field. Since the success of existing audio classification methods is based on labor-intensive acquisition of high-quality audio labels, inspired by the mature few-shot image recognition methods [34, 42, 43, 46], few-shot audio recognition [24, 33, 35] has received attention in data-limited scenarios, where models are constrained to learn from limited audio with labels. Early metric-based few-shot audio recognition methods [5, 16, 17, 23, 33, 35, 37] aimed to cluster unlabeled examples based on their distance from several examples of each category. Among them, the matching network [35] learns a set of mapping matching relationships between supporting data and queries. The prototype network [33] learns different category prototype representations and maps categories based on the Euclidean distance between the query representation and the prototype representation.

However, without the support of multi-modal basic models [10, 11, 15, 38], it is difficult for early methods to perfectly solve the problem of few-shot audio recognition. Recently, inspired by the success of Tip-adapter [43] in image recognition, Treff-adapter [24] conducts coarse-grained correspondence learning through efficient adapters to guide the audio-language model to adapt to few-shot Audio Recognition. Specifically, Treff-adapter learns the coarse-grained correlation between the test audio representation and the audio representation of the known category, and utilizes the key-value cache to map the category of the test audio. And the zero-shot cross-modal contrast is performed to capture the coarse-grained correlation between the category name representation and the coarse-grained audio representation. Although the above methods achieve good performance, their coarse-grained correspondence learning makes it difficult to establish multi-grained matching patterns between audio and category concepts. Therefore, we propose multi-grained correspondence learning for effectively learning multi-grained matching patterns between multi-level audio information and diverse category characteristics, which benefits from our application of optimal transport for fine-grained correspondence learning.

## 2.3 Optimal Transport

Optimal transport [29] was originally developed to solve the problem of how to move multiple materials at the same time at the lowest cost, and can be used to evaluate the distance between two probability distributions. Since many modern statistical and machine learning problems can be reformulated as finding optimal transport graphs between two probability distributions, optimal transport has recently attracted widespread attention in the fields of machine learning and computer vision, such as document matching [21, 40], image matching [1, 26, 36, 41, 45], video retrieval [25], and visual-language models [4, 8, 19, 31, 39]. However, none of these works specifically focus on few-shot audio recognition and correspondence learning between audio and text, which is the main focus of our research. Experimental results show that fine-grained correspondence learning based on optimal transport effectively establishes flexible fine-grained matching patterns between audio and category concepts.

## 3 METHODOLOGY

In this section, we elaborate on each component of our proposed method, whose overall structure is shown in Fig.2. We first introduce how to learn multi-modal representations with the assistance of generative models in Sec.3.1. Subsequently, we detailed how to promote multi-grained correspondence learning through multi-grained key-value cache and multi-grained cross-modal contrast in Sec.3.2 and Sec.3.3. Finally, we describe in Sec.3.4 the evaluation of the overall matching probability for few-shot recognition.

## 3.1 Gen-assisted multi-modal representation

*3.1.1 Audio representation.* For a sampled audio, the mel spectrogram of the audio is input into the CNN14 audio encoder [2] to extract the audio feature map $F^a \in \mathbb{R}^{n_a \times f_a \times d}$, where $n_a/f_a$ represent the time/frequency dimension and $d$ is the latent feature dimension. The width and height of an audio mel spectrogram represent different information (i.e. time and frequency bins). Since the length of time is usually much longer than the length of frequency bins, we perform average pooling on the audio feature map $F^a$ along the frequency direction to obtain a fine-grained audio representation (clip-level) $R_a^f = \{a_i\}_{i=1}^{n_a} \in \mathbb{R}^{n_a \times d}$, where $n_a$ represents the number of time segments. The average of the fine-grained audio representation $R_a^f$ can be utilized as the coarse-grained audio representation (overall-level) $R_a^c \in \mathbb{R}^d$.

*3.1.2 Audio caption representation.* We noticed that in real-world scenarios, audio clips on video sharing platforms often come with relevant textual descriptions, such as titles and tags. The semantic information contained in these texts is crucial for improving audio recognition in situations where data availability is limited. This inspires us to utilize pre-trained encoder-decoder models [28] to generate audio captions without training. Specifically, we adopt HTSAT [2] as the audio encoder to extract audio features. The HTSAT audio encoder is an audio transformer similar to the visual transformer (VIT). And we use BART [22] as the language decoder to generate captions based on the audio features extracted from the encoder. From this, we generate a corresponding caption for each audio, and BERT [7] is used as a text encoder to extract the audio caption representation (cap-level) $R_t \in \mathbb{R}^d$.

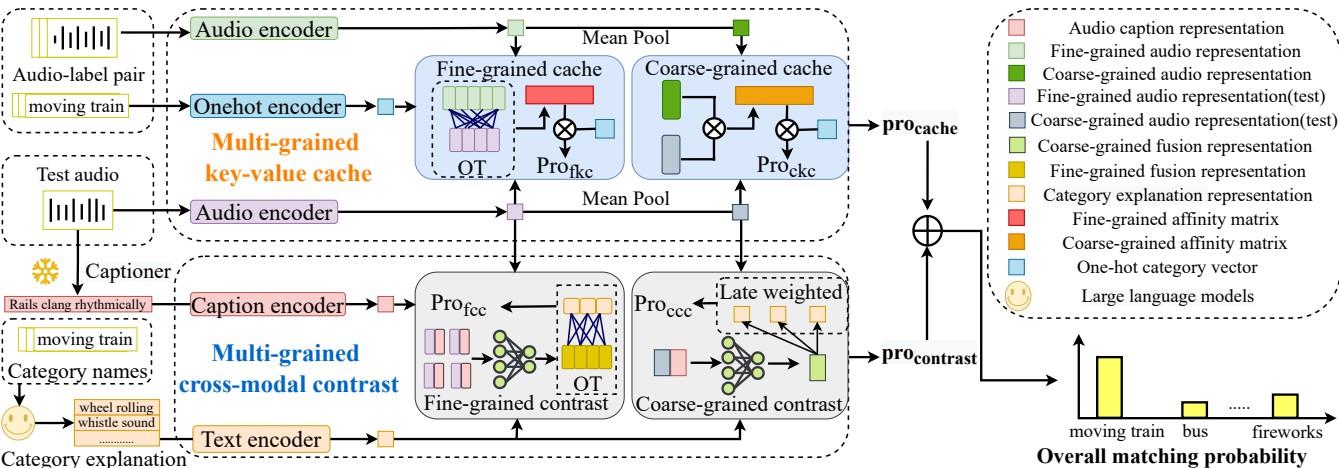

**Figure 2: Illustration of the proposed multi-grained correspondence learning, which is used to guide the adaptation of audio-language models for few-shot audio recognition. Multi-modal representation learning with the assistance of generative models is performed to mine the multi-level information of audio and the diverse characteristics of category concepts. Multi-grained key-value cache and multi-grained cross-modal contrast are then proposed to effectively learn multi-grained matching patterns between audio and category concepts through optimal transport modeling (OT) and late weighted fusion.**

*3.1.3 Category representation.* Traditional audio recognition methods map category labels into discrete numerical codes for classification, which ignores the semantic information contained in category names. We use preset prompts to guide large language models (e.g. chatgpt[1]) to generate descriptive explanations of category labels, which are used to understand the potential meaning of category concepts. Specifically, we carefully design a variety of prompt templates to guide chatgpt to generate $n_t$ descriptive explanations for each category label. Different category explanations illustrate different discriminative characteristics of audio in that category. Since BERT [7] has strong language modeling capabilities, it is used as a text encoder to extract text representations of descriptive explanations $R_e = \{e_i\}_{i=1}^{n_e} \in \mathbb{R}^{n_e \times d}$, where $n_e$ represents the number of explanations corresponding to each category.

## 3.2 Multi-grained key-value cache

Multi-grained key-value cache is designed to capture multi-grained correlations between test audio and known category audio to map the category of the test audio. We consider key-value cache forms of multi-grained audio representations, including coarse-grained key-value cache and fine-grained key-value cache.

*3.2.1 Coarse-grained key-value cache.* The coarse-grained key-value cache is used to map the corresponding categories by mining the similarity between the coarse-grained representations of the test audio and training audios. Under the "K-shot N-class" few-shot recognition setting, audio needs to be divided into $N$ categories, and each category provides $K$ audio samples of known categories. For each audio $a_i$ in the $NK$ training sets, the audio representations are treated as keys and their corresponding categories are treated as values after being converted into N-dimensional one-hot vectors

$V_i \in \mathbb{R}^{1 \times N}$. And the representations of test audio of unknown categories are treated as queries. The similarity between the query and the key can be regarded as the affinity between the query and the value, which is used to map the matching probability between the query audio and $N$ categories.

Specifically, for coarse-grained key-value cache, we consider coarse-grained representations of a test audio $R_{a,test}^c \in \mathbb{R}^d$ and training audios $R_{a,train}^{c'} = \{R_{a,i}^{c'}\}_{i=1}^{NK} \in \mathbb{R}^{NK \times d}$. Inspired by Treff-adapter [24], cosine similarity is used to evaluate the coarse-grained affinity $M_{aff,i}^c$ between the test audio and the training audio set. Finally, the matching probability $Pro_{ckc} \in \mathbb{R}^{1 \times N}$ between the test audio and $N$ categories is determined by the coarse-grained affinity and the one-hot category vector $V_c = \{V_i\}_{i=1}^{NK} \in \mathbb{R}^{NK \times N}$, defined as follows:

$$M_{aff,i}^c = \varphi((R_{a,test}^c)(R_{a,i}^{c'})^T) \tag{1}$$

$$Pro_{ckc} = \sum_{i=1}^{NK} M_{aff,i}^c V_i \tag{2}$$

where $\varphi(x) = exp(-\beta(1-x)))$ represents the scaling function.

*3.2.2 Fine-grained key-value cache.* The fine-grained key-value cache is used to map the corresponding categories by mining the similarity between the fine-grained representations of the test and training audios. However, as explained in Sec.1, there is a temporal misalignment problem between the test audio and the training audios. The key information in audio is distributed randomly over time, and it can appear at the beginning or end of the audio, so the relevant parts of two similar audios may not appear in the same time segment (Fig.1-a). Since fine-grained audio representations encompass audio information across multiple segments in chronological order, We need a metric that is independent of segment

[1]https://openai.com/blog/chatgpt/

order to establish flexible correspondences between fine-grained representations of test and training audios.

Therefore, we propose to apply optimal transport (OT) to solve the temporal misalignment problem by modeling the empirical distribution of fine-grained audio representations and learning a transport plan to capture the fine-grained flexible correspondence of test and training audios. This is because the optimal transport (OT) distance, as a metric for distribution comparison, can establish a flexible correspondence (regardless of the order) between sampling points of different probability distributions. This allows optimal transport to align relevant audio information at different time periods between audios without being constrained by time order. Specifically, taking a test audio sample and a training audio sample as an example, we apply two empirical distributions $P$ and $Q$ to model the fine-grained audio representation of a test sample $R_a^f = \{a_i\}_{i=1}^{n_a} \in \mathbb{R}^{n_a \times d}$ and fine-grained audio representation of a training sample $R_a^{f'} = \{a_j'\}_{j=1}^{n_a'} \in \mathbb{R}^{n_a' \times d}$.

$$P = \sum_{i=1}^{n_a} p_i \delta_{a_i} \quad , \sum_{i=1}^{n_a} p_i = 1 \qquad (3)$$

$$Q = \sum_{j=1}^{n_a'} q_j \delta_{a_i'} \quad , \sum_{j=1}^{n_a'} q_j = 1 \qquad (4)$$

where $\delta_t$ represents the dirac function defined at $t$, and the discrete probability vectors $p = \{p_i\}_{i=1}^{n_a}$ and $q = \{q_j\}_{j=1}^{n_a'}$ belong to $n_a$ and $n_a'$-dimensional simplex. Optimal transport aims to minimize the OT distance between distributions $P$ and $Q$. This is used to capture flexible correlation patterns between test audio and training audios without temporal order constraints, which can be formulated as an optimization problem guided by entropy regularization:

$$d_{OT}^\lambda(P, Q; C) = min \sum_{i=1}^{n_a} \sum_{j=1}^{n_a'} T_{ij} C_{ij} - \lambda h(T) \qquad (5)$$

$$T\mathbf{1}_{n_a'} = p \quad T^T \mathbf{1}_{n_a} = q \qquad (6)$$

$$c_{ij} = 1 - \frac{a_i(a_j')^T}{||a_i||_2 ||a_j'||_2} \qquad (7)$$

Among them, $T \in \mathbb{R}^{n_a \times n_a'}$ is the transport plan to be learned, $T_{ij}$ represents the transport probability from the i-th audio segment to the j-th audio segment. The larger its value, the closer the semantic connection between the two segments. $h(T)$ is the entropy constraint, and $\lambda \geq 0$ is the weight of entropy regularization. $C_{ij}$ is the transport cost between the i-th audio segment and the j-th audio segment. Considering that the greater the transport cost, the lower the transport probability, the cosine similarity between audio segments is used to construct the cost matrix $C \in \mathbb{R}^{n_a \times n_a'}$. And we consider uniform distribution for $p$ and $q$. Then, we can obtain a fast optimized matrix solution $T^*$ for the transport plan $T$ through Sinkhorn fixed point iteration [6]:

$$T^* = Diag(k_1)exp(-C/\epsilon)Diag(k_2) \qquad (8)$$

$$Iteratively : p/(exp(-C/\epsilon)k_2)-> k_1, q/(exp(-C/\epsilon)^T k_1)-> k_2$$

Among them, $k_1$ and $k_2$ are predefined left and right scaling vectors. During the above optimization process, we obtain the optimal transport plan $T^* \in \mathbb{R}^{n_a \times n_a'}$ between fine-grained representations of test and training audios, which reflects the tightness of the semantic connection between audio segments. Accordingly, the optimal transport plan is used to define the fine-grained similarity $Sim_{OT}$ between test audio representation $R^f$ and training audio representation $R^{f'}$ as follows:

$$Sim_{OT}(R^f, R^{f'}) = \sum_{i=1}^{n_a} \sum_{j=1}^{n_a'} T_{ij}^* \frac{a_i(a_j')^T}{||a_i||_2 ||a_j'||_2} \qquad (9)$$

Similar to coarse-grained, for fine-grained key-value cache, we consider fine-grained representations of a test audio $R_{a,test}^f \in \mathbb{R}^{n_a \times d}$ and $NK$ training audios $R_{a,train}^{f'} = \{R_{a,i}^{f'}\}_{i=1}^{NK}$. The fine-grained similarity $Sim_{OT}$ is used to evaluate the fine-grained affinity $M_{aff,i}^f$ between the test audio $R_{a,test}^f$ and the training audios $R_{a,i}^{f'} \in \mathbb{R}^{n_{a,i}' \times d}$. Finally, the matching probability $Pro_{fkc} \in \mathbb{R}^{1 \times N}$ between the test audio and $N$ categories is defined as follows:

$$M_{aff,i}^f = \varphi(Sim_{OT}(R_{a,test}^f, R_{a,i}^{f'})) \qquad (10)$$

$$Pro_{fkc} = \sum_{i=1}^{NK} M_{aff,i}^f V_i \qquad (11)$$

In summary, the overall matching probability of the multi-grained key-value cache module can be defined as follows:

$$Pro_{cache} = Pro_{ckc} + Pro_{fkc} \qquad (12)$$

## 3.3 Multi-grained cross-modal contrast

Multi-grained cross-modal contrast aims to mine multi-grained correlations between audio and category explanations assisted by audio captions. We explore the fusion method of multi-grained audio representation and caption representation, and consider the correlation form of multi-grained fusion representation and diverse category explanation representation, including fine-grained cross-modal contrast and coarse-grained cross-modal contrast.

*3.3.1 Coarse-grained cross-modal contrast.* Coarse-grained cross-modal contrast aims to mine coarse-grained similarity between audio and category explanations. We first perform representation interaction between coarse-grained audio representation and caption representation to obtain a semantically enhanced coarse-grained fusion representation. Our motivation is that although captions and audio are presented in different modal forms, they are essentially expressions of audio information. Captions tend to focus on salient concepts but miss out on details, while audio retains the complete message but is interfered with by redundant segments. A natural idea is to encourage cross-modal interaction of captions with audio to learn complementary patterns of both. Specifically, after the coarse-grained audio representation $R_a^c \in \mathbb{R}^d$ and caption representation $R_t \in \mathbb{R}^d$ are spliced, linear transformation is used to project them into a new embedding space as a coarse-grained fusion representation $R_{fu}^c \in \mathbb{R}^d$, which is defined as follows:

$$R_{fu}^c = W_c(concat(R_a^c, R_t)) \qquad (13)$$

where $W_c \in \mathbb{R}^{2d \times d}$ represents the learnable linear layer. Then, we propose a late weighted fusion method (LW) to guide the matching of coarse-grained fusion representations $R_{fu}^c \in \mathbb{R}^d$ with diverse category explanation representations $R_e = \{e_i\}_{i=1}^{n_e} \in \mathbb{R}^{n_e \times d}$. Our motivation is that different category explanations illustrate different characteristics of audio in this category, but even if the audio belongs to the same category, the significance of different characteristics will be different. Therefore, we propose a late weighted fusion method to guide the model to focus on the correlation between audio and salient discriminative characteristics. Specifically, we consider the cosine similarity $sim_i^c$ between the coarse-grained fusion representation $R_{fu}^c \in \mathbb{R}^d$ and the category explanation representation $e_i \in \mathbb{R}^d$, and adaptively obtain the characteristic saliency weight $w_i$ according to the correlation between the audio and different characteristics. The coarse-grained similarity $Sim_{LW}^c$ between coarse-grained fusion representation $R_{fu}^c \in \mathbb{R}^d$ and diverse category explanation representation $R_e = \{e_i\}_{i=1}^{n_e} \in \mathbb{R}^{n_e \times d}$ can be defined as follows:

$$sim_i^c = \frac{R_{fu}^c(e_i)^T}{||R_{fu}^c||_2||e_i||_2}, w_i = \frac{e^{sim_i^c}}{\sum_{i=1}^{n_e} e^{sim_i^c}} \tag{14}$$

$$Sim_{LW}^c(R_{fu}^c, R_e) = \sum_{i=1}^{n_e} w_i sim_i^c \tag{15}$$

From this, the matching probability $Pro_{ccc} = \{Pro_{ccc,j}\}_{j=1}^N \in \mathbb{R}^{1 \times N}$ between the test audio and $N$ categories can be defined as follows:

$$Pro_{ccc,j} = Sim_{LW}^c(R_{fu}^c, R_{e,j}) \tag{16}$$

Among them, $R_{e,j}$ represents the diverse category explanation representation of the j-th category.

*3.3.2 Fine-grained cross-modal contrast.* Fine-grained cross-modal contrast aims to mine fine-grained similarity between audio and category explanations. We first perform representation interaction between fine-grained audio representation $R_a^f = \{a_i\}_{i=1}^{n_a} \in \mathbb{R}^{n_a \times d}$ and caption representation $R_t \in \mathbb{R}^d$ to obtain semantically enhanced fine-grained fusion representation $R_{fu}^f = \{f_i\}_{i=1}^{n_a} \in \mathbb{R}^{n_a \times d}$. Specifically, after each segment representation is spliced with the caption representation, the learnable learnable linear layers $W_f \in \mathbb{R}^{2d \times d}, W_l \in \mathbb{R}^{d \times d}$ are used to project it into a new embedding space as a fine-grained fusion representation $R_{fu}^f$, which is defined as follows:

$$f_i = W_l W_f(concat(a_i, R_t)) \tag{17}$$

However, as explained in Sec.1, we noticed that there is a semantic intersection problem between audio and category explanations, that is, some segments in the audio can express multiple semantics at the same time (Fig.1-b). Since optimal transportation can establish flexible associations between sampling points of different probability distributions (e.g. one-to-many, many-to-many), we propose to apply optimal transportation (OT) to solve the semantic intersection problem. Specifically, similar to Sec.3.2.2, we first conduct empirical distribution modeling of fine-grained fusion representations $R_{fu}^f = \{f_i\}_{i=1}^{n_a} \in \mathbb{R}^{n_a \times d}$ and diverse category interpretation

representations $R_e = \{e_i\}_{i=1}^{n_e} \in \mathbb{R}^{n_e \times d}$. We then facilitate flexible fine-grained correspondences between audio and corresponding category explanations by minimizing the optimal transport (OT) distance between distributions. After an optimization process similar to Eq.8, the fine-grained similarity $Sim_{OT}^f$ between the fine-grained fusion representation $R_{fu}^f$ and the diverse category explanation representation $R_e$ is defined as follows:

$$Sim_{OT}^f(R_{fu}^f, R_e) = \sum_{i=1}^{n_a} \sum_{j=1}^{n_e} T_{ij}'^* \frac{f_i(e_j)^T}{||f_i||_2||e_j||_2} \tag{18}$$

The fast optimization matrix solution $T_{ij}'^*$ of the optimal transport plan represents the closeness of the semantic connection from the i-th audio segment to the j-th category explanation. From this, the matching probability $Pro_{fcc} = \{Pro_{fcc,j}\}_{j=1}^N \in \mathbb{R}^{1 \times N}$ between the test audio and $N$ categories can be defined as follows:

$$Pro_{fcc,j} = Sim_{OT}^f(R_{fu}^f, R_{e,j}) \tag{19}$$

In summary, the overall matching probability of the multi-grained cross-modal contrast module can be defined as follows:

$$Pro_{contrast} = Pro_{ccc} + Pro_{fcc} \tag{20}$$

### 3.4 Overall matching probability evaluation

The overall matching similarity is jointly evaluated by the two modules of multi-grained key-value cache and multi-grained cross-modal contrast, and is defined as follows:

$$pro_{total} = \alpha Pro_{cache} + Pro_{contrast} \tag{21}$$

Among them, $Pro_{total} \in \mathbb{R}^{1 \times N}$ represents the overall matching probability between a piece of test audio and the $N$ categories to be classified, and $\alpha$ represents the hyperparameter that adjusts the effect of multi-grained key-value cache on the overall matching probability. And we predict the audio category by choosing the category with the highest probability. Cross-entropy loss is used for optimization during training:

$$\zeta(\theta) = -\frac{1}{B} \sum_{i=1}^B \sum_{j=1}^N y_{i,j} log \hat{y}_{i,j} \tag{22}$$

where $B$ is the total number of training examples.If the j-th category of the i-th sample corresponds to the true category label, then $y_{i,j} = 1$, otherwise $y_{i,j} = 0$. $\hat{y}_{i,j}$ represents the matching probability between the i-th audio sample and the j-th category, which can be defined as the j-th element of $Pro_{total}$ corresponding to the i-th sample. $\theta = \{W_c, W_l, W_f\}$ represents learnable parameters.

## 4 EXPERIMENTS

### 4.1 Datasets

**ESC-50** The ESC-50 dataset [30] is a widely used audio recognition dataset. It consists of 2,000 environmental recording clips, each being 5 seconds long, and has 50 semantic category labels. There are 40 sample recordings for each category. For evaluation purposes, the label set is repeatedly sampled into 5 subsets, with 15 label classes in each subset, enabling 5-fold cross-validation, following the setting in Treff-adapter [24].

**Table 1: Comparison of audio classification accuracy (%) with other state-of-the-art methods on ESC-50 dataset and FSDkaggle18 dataset under different training sample settings. Our proposed multi-grained correspondence learning method consistently outperforms all compared methods in all settings. w/o CLAP: Non-CLAP backbone model, CLAP-base: Basic CLAP model, CLAP-adapt: CLAP model adapted for audio recognition.**

| Dataset | Type | Method | 5-way Acc | | 12-way Acc | | Avg |
|---|---|---|---|---|---|---|---|
| | | | 1-shot | 10-shot | 1-shot | 10-shot | |
| ESC-50 | w/o CLAP | MatchNet(2016) [35] | - | 86.83 | - | 71.81 | - |
| | w/o CLAP | ProtoNet(2017) [33] | - | 88.18 | - | 77.70 | - |
| | CLAP-base | CLAP-zeroshot(2022) [10] | 96.84 | 96.84 | 93.65 | 93.65 | 95.25 |
| | CLAP-adapt | Tip-adapter-F(2022) [43] | 96.97 | 97.52 | 93.97 | 95.58 | 96.01 |
| | CLAP-adapt | Treff-adapter(2023) [24] | 97.44 | 98.53 | 94.83 | 96.29 | 96.77 |
| | CLAP-adapt | Ours | **98.63** | **99.54** | **98.42** | **99.51** | **99.03** |
| FSDkaggle18 | CLAP-base | CLAP-zeroshot(2022) [10] | 88.76 | 88.76 | 80.24 | 80.24 | 84.50 |
| | CLAP-adapt | Tip-adapter-F(2022) [43] | 88.98 | 90.75 | 80.83 | 83.92 | 86.12 |
| | CLAP-adapt | Treff-adapter(2023) [24] | 89.37 | 92.31 | 81.58 | 86.94 | 87.55 |
| | CLAP-adapt | Ours | **90.67** | **94.88** | **88.06** | **93.33** | **91.74** |

**FSDkaggle18** The FSDkaggle18 dataset [12] contains 11,073 audio clips with durations ranging from 300ms to 30s. Each clip is annotated with one of the 41 labels from the AudioSet Ontology [13]. All audio samples have a corresponding label. Similar to ESC-50, the class label set is repeatedly sampled into 5 subsets, each with 10 label classes, for 5-fold cross-validation, following the data segmentation method used in Treff-adapter [24].

## 4.2 Experimental Settings

Our framework selects the audio-language model CLAP [10] as the backbone, in which CNN14 [2] is selected as the audio encoder backbone and BERT [7] is selected as the text encoder. Pre-trained weights from CLAP are loaded by the model and frozen during training. For audio data, we adopt the same data preprocessing method as Treff-adapter [24], and use the pre-trained encoder-decoder model [28] to generate a caption for each audio. For category names, based on our preset multiple prompt templates (e.g. what are the characteristics of the sound of [class]?), we use the api provided by chatgpt to generate 8 category explanations for each category name. During the training phase, the training epochs are set to 20, $\beta$ is set to 5, and $\alpha$ is set to 1.1. The optimizer is Adam [20] and the learning rate is set to 10-4. And like previous studies on few-shot audio recognition [24, 43], we adopt the "N-way K-shot" setting. In this case, $N$ categories are randomly selected for classification, and each category contains $K$ training samples. During the testing phase, we performed five-fold cross-validation and evaluated the model performance as an accuracy metric. All experiments are performed on a single 48GB A40 GPU based on the pytorch framework.

## 4.3 Comparison Experiments

**Main Results** Tab.1 presents a comparative analysis of few-shot audio recognition results achieved by our proposed framework against other leading methods on the ESC-50 and FSDkaggle18 datasets. Notably, methods leveraging audio-language models (CLAP) generally surpass metric learning techniques [33, 35], emphasizing the significance of guiding audio-language models to adapt to downstream audio recognition tasks. Among the various CLAP-adapt methods [24, 43], our approach excels in comparison, delivering

superior performance under diverse shot settings. Furthermore, as the number of training samples increases, our method exhibits a commensurate enhancement in performance, highlighting its potential to effectively facilitate the adaptation of audio-language models to downstream audio recognition tasks.

In contrast to previous approaches, such as the state-of-the-art Treff-adapter [24], which relies solely on audio data from limited training sets for coarse-grained learning, our framework offers a more comprehensive and nuanced approach to audio recognition. As a testament to its effectiveness, our framework has demonstrated reliable recognition accuracy across multiple shots settings in the widely used ESC-50 dataset. Notably, we have achieved a significant improvement of 2.26% in recognition performance compared to Treff-adapter. Moreover, in the more challenging FSDkaggle18 dataset, our framework has achieved a remarkable recognition accuracy of 91.74%, surpassing the previous best method by an absolute improvement of 4.19%.

Furthermore, it is noteworthy that our proposed method significantly enhances the performance of zero-shot CLAP [10] when guided by a limited amount of training data. Remarkably, we observed absolute improvements of 3.78% and 7.24% on the ESC-50 dataset and FSDkaggle18 dataset respectively. These results clearly demonstrate that our method effectively boosts the adaptability of the audio-language model to downstream audio recognition datasets, making it a highly versatile and effective approach.

## 4.4 Ablation Experiments

In this section, we conduct comprehensive ablation experiments to elucidate the impact of each component in our proposed method. All ablation experiments were performed on the FSDkaggle18 dataset. **Effect of generated audio captions** To delve deeply into the influence of captions on multi-grained correspondence learning, we meticulously crafted three models: a caption-solely dependent model, an audio-solely dependent model, and a model that integrates multi-modal information. As exhibited in Tab.2, it is challenging to attain reliable audio recognition solely relying on information from a single modality. Conversely, the integration of multi-modal information enhances the caption-only model by 10.18% and the

audio-only model by 2.81%. This is attributed to the fact that captions and audio present audio information at different levels and in distinct modal forms. Captions tend to concentrate on prominent concepts but overlook finer details, whereas audio preserves the entire message yet is often clouded by redundant segments. The multi-modal representation interaction discussed in Sec.3.3 effectively learns the complementary nature between captions and audio, resulting in a semantically enriched fusion representation.

**Table 2: Ablation experimental results for generated audio captions. Caption-only: Audio recognition using only captions, Audio-only: Audio recognition using only audios.**

| Model | 5-way Acc | | 12-way Acc | | Avg |
|---|---|---|---|---|---|
| | 1-shot | 10-shot | 1-shot | 10-shot | |
| Caption Only | 85.04 | 85.87 | 77.16 | 78.17 | 81.56 |
| Audio Only | 90.23 | 93.38 | 83.86 | 88.25 | 88.93 |
| Caption+Audio | **90.67** | **94.88** | **88.06** | **93.33** | **91.74** |

**Effect of generated category explanations** As shown in Tab.3, we discuss the impact of the number of generated category explanations on model performance and compare the performance of models using category names instead of category explanations. We observe that models that rely solely on category name guidance do not perform well. However, as the number of category explanations gradually increases, the model performance shows an improvement trend. This should be attributed to the fact that rich category explanations illustrate the diverse characteristics of category concepts and effectively capture the potential meaning of category concepts. Considering the quality of text generated by large language models, generating 8 corresponding category explanations for each category is an efficient choice.

**Table 3: Ablation experimental results for generated category explanations. Category name: Model based on category name, Explain: Model based on category explanations.**

| Model | 5-way Acc | | 12-way Acc | | Avg |
|---|---|---|---|---|---|
| | 1-shot | 10-shot | 1-shot | 10-shot | |
| Category Name | 89.92 | 94.07 | 87.18 | 92.43 | 90.90 |
| Explain ($n_e = 2$) | 89.92 | 94.49 | 87.53 | 92.81 | 91.19 |
| Explain ($n_e = 4$) | 90.53 | 94.63 | 87.50 | 93.15 | 91.45 |
| Explain ($n_e = 6$) | 90.62 | 94.75 | 87.91 | 93.30 | 91.65 |
| Explain ($n_e = 8$) | **90.67** | **94.88** | **88.06** | **93.33** | **91.74** |

**Effect of multi-grained key-value cache** As shown in Tab.4, we examine the impact of the multi-grained key-value cache module and its components on model performance. We observe that the multi-grained key-value cache module achieves an absolute improvement of 1.17%, while the coarse-grained and fine-grained key-value caches achieve an absolute improvement of 0.35% and 0.45% respectively. This is because key-value caches of different granularities capture diverse correlations between audios and promote each other to achieve stronger recognition capabilities.
**Effect of multi-grained cross-modal contrast** As shown in Tab.5, we explored the impact of the multi-grained cross-modal contrast

**Table 4: Ablation experimental results for multi-grained key-value cache. $Pro_{cache}$: Matching probability of the multi-grained key-value cache module, $Pro_{ckc}$: Matching probability of the coarse-grained key-value cache, $Pro_{fkc}$: Matching probability of the fine-grained key-value cache, w/o: Remove this module from the entire model.**

| Model | 5-way Acc | | 12-way Acc | | Avg |
|---|---|---|---|---|---|
| | 1-shot | 10-shot | 1-shot | 10-shot | |
| Total | **90.67** | **94.88** | **88.06** | **93.33** | **91.74** |
| w/o $Pro_{ckc}$ | 90.49 | 94.11 | 87.83 | 93.14 | 91.39 |
| w/o $Pro_{fkc}$ | 90.01 | 94.43 | 87.41 | 93.29 | 91.29 |
| w/o $Pro_{cache}$ | 89.72 | 93.08 | 86.96 | 92.53 | 90.57 |

module and its components on model performance. Among them, the multi-grained cross-modal contrast module achieves an absolute improvement of 12.07%, while the coarse-grained/fine-grained cross-modal contrast achieves an absolute improvement of 5.22% and 0.74% respectively. This may be due to the fact that each granularity of cross-modal contrast plays a different role in the audio recognition task, and the different granularities of cross-modal contrast can promote each other.

**Table 5: Ablation experimental results for multi-grained cross-modal contrast. $Pro_{contrast}$: Matching probability of the multi-grained cross-modal contrast module, $Pro_{ccc}$: Matching probability of the coarse-grained cross-modal contrast, $Pro_{fcc}$: Matching probability of the fine-grained cross-modal contrast, w/o: Remove this module from the entire model.**

| Model | 5-way Acc | | 12-way Acc | | Avg |
|---|---|---|---|---|---|
| | 1-shot | 10-shot | 1-shot | 10-shot | |
| Total | **90.67** | **94.88** | **88.06** | **93.33** | **91.74** |
| w/o $Pro_{fcc}$ | 89.54 | 94.18 | 87.43 | 92.85 | 91.00 |
| w/o $Pro_{ccc}$ | 85.44 | 94.32 | 76.57 | 89.73 | 86.52 |
| w/o $Pro_{contrast}$ | 75.36 | 92.57 | 63.59 | 87.16 | 79.67 |

## 5 Conclusion

In this paper, we propose multi-grained correspondence learning for bootstrapping audio-language models to improve audio recognition with few training samples. Multi-modal representation learning assisted by generative modeling is proposed for extracting potential meanings of multi-level audio information and category concepts. Multi-grained key-value cache and multi-grained cross-modal contrast are proposed to establish flexible correspondences between multi-level audio information and diverse discriminative characteristics through optimal transport modeling and late weighted fusion. Extensive experiments validate the effectiveness of our proposed multi-grained correspondence learning.

## 6 Acknowledgments

This work was supported by the National Science and Technology Major Project of China under Grant No. 2020AAA0108102, the National Natural Science Foundation of China under Grant Nos. 62327808 and 62088102.

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
