# OpenReview forum: "Multi-grained Correspondence Learning of Audio-language Models for Few-shot Audio Recognition"
_acmmm.org/ACMMM/2024/Conference — MM2024 Poster_

### Official Review · Reviewer_8tzz · 2024-05-23

**Rating:** 4
**Confidence:** 3

**Summary:**

To improve few-shot audio recognition, the authors proposed multi-grained correspondence learning for bootstrapping audio-language models. Existing methods often struggle with capturing intricate matching patterns between audio and category concepts. The proposed method addresses this by enriching multimodal representation learning through generative models, establishing multi-grained matching patterns via key-value cache and cross-modal contrast. Additionally, the authors integrate optimal transport to handle temporal misalignment and semantic intersection issues, enabling flexible fine-grained matching. Experimental results on ESC-50 and FSDkaggle18 datasets demonstrate the effectiveness of the proposed approach, achieving state-of-the-art performance.

**Strengths:**

- The authors have a good point of view, recognizing the significant challenges in acoustic signals shown in Figure 1, namely temporal misalignment and semantic intersection. To address these challenges, the proposed method based on multi-grained correspondence learning with optimal transport is very novel.
- From Table 1, the effectiveness of the proposed method is demonstrated by comparing it with state-of-the-art methods using two different datasets.

**Limitations:**

- Although the classification accuracy shows the effectiveness of the proposed method, it does not indicate whether the problems (a) and (b) in Figure 1 have been solved. It also does not indicate the limitations of the proposed method.
- It is not clear what the authors want to show in Table 2. What are "a caption-solely dependent model", "an audio-solely dependent model" and "a model that integrates multi-modal information"?
- The effectiveness of optimal transport should be verified. Couldn't a simpler distance be used?
- In Table 3, what happens if n_e, the number of explanations corresponding to each category, is set to 9 or over?
- Preset prompts should be indicated. It was difficult to imagine what descriptive explanations of category labels generated using the large language model would look like.

**Suitability:**

3

---

### Official Review · Reviewer_rH2A · 2024-05-24

**Rating:** 4
**Confidence:** 3

**Summary:**

The paper introduces a novel approach to few-shot audio recognition by leveraging multi-grained
correspondence learning. This method aims to improve the adaptability of audio-language models
to audio recognition tasks with limited training samples through generative model-assisted multimodal
representation learning and optimal transport for fine-grained matching.

**Strengths:**

This paper has these strengths:
1) The paper proposes a significant advancement in few-shot audio recognition by focusing on the
intricate patterns between audio and category concepts, the idea is novel.
2) The concept of multi-grained correspondence learning is theoretically sound and provides a
more nuanced approach to capturing the complexities of audio data compared to coarse-grained
methods.
3) The use of optimal transport to address temporal misalignment and semantic intersection is a
clever application of OT, demonstrating both effectiveness and innovation.
4) The paper presents some experimental results to show the effectiveness of the proposed method,
achieving state-of-the-art results on ESC-50 and FSDkaggle18 datasets.

**Limitations:**

1) The experiment should be conducted on more and larger datasets to show its generality.
2 )While the paper mentions state-of-the-art results, a more detailed comparison with the
existing state-of-the-art methods, including potential trade-offs, would be valuable.
Also, more methods like [1] that incorporate LLM for few-shot and zero-shot recognition
should be discussed or compared in the experiment part.

[1]Liang J, Liu X, Wang W, et al. Acoustic Prompt Tuning: Empowering Large
Language Models with Audition Capabilities[J]. 2023.

**Suitability:**

3

---

### Official Review · Reviewer_G6fD · 2024-05-24

**Rating:** 3
**Confidence:** 3

**Summary:**

The paper introduces a novel approach to enhance audio recognition capabilities with limited training samples. The core of the methodology lies in multi-grained correspondence learning, which leverages multi-modal representation learning aided by generative models. This approach is designed to extract meaningful insights from various levels of audio information and category concepts. The paper proposes the use of multi-grained key-value cache and multi-grained cross-modal contrast mechanisms to establish flexible correspondences between different levels of audio information and the discriminative characteristics of these sounds. These mechanisms rely on optimal transport modeling and late-weighted fusion for efficiency. To validate the effectiveness of their proposed method, the authors conducted experiments on two benchmark datasets for few-shot audio recognition, ESC-50, and FSDkaggle18, achieving state-of-the-art results.

**Strengths:**

1. The paper introduces a novel multi-grained correspondence learning framework that effectively leverages multi-modal representation learning and generative models. This innovative approach addresses the challenge of few-shot audio recognition by mining multi-level audio information and category concepts.
2. The technical aspects of the proposed methods, including the multi-grained key-value cache and multi-grained cross-modal contrast, are well-founded. These methods are based on principles including optimal transport modeling and late weighted fusion, which contribute to their effectiveness.
3. The authors provide an empirical evaluation of their approach on two benchmark datasets, ESC-50 and FSDkaggle18. The state-of-the-art results achieved by their method serve as strong evidence of its effectiveness and applicability to few-shot audio recognition tasks.

**Limitations:**

1. While the paper presents impressive results on two benchmark datasets, the scope of these datasets is somewhat limited(only two benchmark datasets).  Has the proposed approach been evaluated on a broader range of datasets, including those with more diverse audio samples and real-world scenarios?
2. The paper could further address the generalizability of the proposed method across different audio recognition tasks beyond the few-shot scenario. Is the approach able to perform in more generalized settings or when applied to other related tasks?
3. When the paper analyzes the effect of multi-grained key-value cache, it is observed that the absolute improvements brought by the coarse-grained and fine-grained key-value cache modules are relatively small, at 0.35% and 0.45% respectively. Does it suggest that the effect of these modules is limited? Designing complex modules for a limited performance improvement is not suggested.
4. The few-shot methods are usually sensitive to the hyper-parameters and introduce larger performance variations. Are the results of the proposed methods statistically significant compared to the baselines?

**Suitability:**

3

---

### Meta-Review · Area_Chair_ywXy · 2024-06-30

**Recommendation:** Accept (Poster)
**Confidence:** 4

**Metareview:**

The authors aim to improve the adaptability of audio-language models for few-shot audio recognition. Although this work is interesting, potential issues should be explained and addressed. More details on the comparison with existing state-of-the-art methods, including potential trade-offs, would be useful. Additionally, more methods that incorporate LLMs for few-shot and zero-shot recognition, such as [1], should be discussed or compared.